# Juvenile Hormone as a contributing factor in establishing midgut microbiota for fecundity and fitness enhancement in adult female *Aedes aegypti*
Mabel L. Taracena-Agarwal [1,2,3,4] ✉, Ana Beatriz Walter-Nuno[1,2], Vanessa Bottino-Rojas[1,2], Alessandra Paola Girard Mejia [4], Kelsey Xu[4], Steven Segal[4], Ellen M. Dotson[3], Pedro L. Oliveira [1,2] & Gabriela O. Paiva-Silva [1,2] ✉

Understanding the factors influencing mosquitoes' fecundity and longevity is important for designing better and more sustainable vector control strategies, as these parameters can impact their vectorial capacity. Here, we address how mating affects midgut growth in *Aedes aegypti*, what role Juvenile Hormone (JH) plays in this process, and how it impacts the mosquito's immune response and microbiota. Our findings reveal that mating and JH induce midgut growth. Additionally, the establishment of a native bacterial population in the midgut due to JH-dependent suppression of the immune response has important reproductive outcomes. Specific downregulation of AMPs with an increase in bacteria abundance in the gut results in increased egg counts and longer lifespans. Overall, these findings provide evidence of a cross-talk between JH response, gut epithelial tissue, cell cycle regulation, and the mechanisms governing the trade-offs between nutrition, immunity, and reproduction at the cellular level in the mosquito gut.

Vector-borne diseases represent a major cause of human morbidity and mortality[1]. The rising global temperatures with socioecological conditions favorable to different vector populations increase the risk of outbreaks of these diseases on all continents, as seen in recent decades[2,3]. Among these illnesses are the ones caused by pathogens transmitted by the *Ae. aegypti* mosquitoes, such as dengue, Zika, chikungunya, and yellow fever diseases, which deserve special attention due to the growing geographical distribution of the vector. The potential concomitant increase in cases substantiates the need for more research focusing on this vector's physiology[4].

*Ae. aegypti*'s vector competence is linked to its hematophagous habit, as the blood intake is central not only for the development of its oocytes and the ability to generate offspring but can also result in the acquisition of the etiological agents they transmit[5]. Newly emerged *Ae. aegypti* females usually feed on sugar-enriched solutions such as floral nectars before finding a blood meal[6]. Amongst the many factors determining female acquisition and use of the blood meal[7,8], mating is one of the more determinant events. Mating effects on previtellogenic[9] and vitellogenic ovary physiology, immunity[10–12]

and behavior in mosquitoes are well documented[13–18]. Male mosquitoes not only transfer sperm to the females, but also deposit a large set of proteins and hormones, such as Juvenile Hormone III (JH) with a wide array of functions[9,19–21]. The contribution of JH, in particular, is interesting due to the pleiotropic effects of this hormone in *Ae. aegypti* adult females. The post-mating increase of systemic JH levels affects ovarian physiology, improving fecundity, prepares the fat body for vitellogenesis and suppresses the immune response mounted by the fat body[22]. In *Drosophila melanogaster*, the midgut epithelium responds to endocrine signaling triggered by the increased titers of JH and 20-Hydroxy-Ecdysone (20HE) after mating to support reproduction[23–25]. In particular, it has been shown that JH signals directly to the midgut epithelium to induce organ enlargement and influences gene expression in order to improve the acquisition and use of nutritional resources for reproduction[23].

The insect gut serves not only as a digestive organ but also as a crucial reservoir for a diverse microbiome[26,27]. This microbiome not only affects the gut epithelium but also influences other organs and even behavior[28–32].

[1]Programa de Biologia Molecular e Biotecnologia, Instituto de Bioquímica Médica Leopoldo de Meis, Universidade Federal do Rio de Janeiro, Rio de Janeiro, Brasil. [2]Instituto Nacional de Ciência e Tecnologia em Entomologia Molecular (INCT-EM), Rio de Janeiro, Brasil. [3]Centers for Disease Control and Prevention (CDC), Atlanta, GA, USA. [4]Entomology Department, Cornell University, College of Agriculture and Life Sciences, Ithaca, NY, USA. ✉e-mail: mlt225@cornell.edu; gosilva@bioqmed.ufrj.br

Therefore, maintaining a delicate balance of immunity is essential to ensure the survival of the indigenous microbiota while protecting the organism from physical and oxidative stresses, toxins, and possible pathogens[33,34]. During this process, epithelial cells may sustain damage, and thus, the integrity of the midgut epithelium relies on continuous cell replenishment through stem cell proliferation and subsequent differentiation[35–38]. In the case of mosquito midguts, it has been observed that biotic and abiotic stress can cause damage and trigger cell division in adult mosquitoes that have either consumed a blood meal or have been infected[34,39–42]. However, while it is well established that newly emerged mosquitoes typically do not feed on blood[43], this has mostly been attributed to sexual immaturity, and the possibility of maturation in the midgut epithelium has not been thoroughly explored yet. Currently, limited information exists regarding the cellular-level maturation process of the adult mosquito midgut and whether it undergoes changes after emergence. If changes do occur in Aedes mosquitoes, these changes could be regulated by endocrine factors, lead to a homeostatic state in adult females, and constitute a maturation process that renders a gut suitable for blood digestion. Recent research has documented that the gut of *Aedes* and *Anopheles* mosquitoes presents an increase in polyploidy during the first hours post-emergence, although the signaling events triggering this event have not yet been characterized[42,44].

Here, we address the effect of mating and JH signaling in the female *Ae. aegypti* midgut, considering the special requirements that an anautogenous reproduction imposes over this organ. Our data shows that mating, and JH in particular, are implicated in the enlargement of the posterior midgut and the establishment of the resident microbiota on it. Our results suggest that immune suppression through downregulation of AMPs expression is necessary for the midgut's microbiome to develop properly. As increased fecundity and longevity after mating had already been described in this organism[13], the data shown here reveal a complex crosstalk between the insect vector and its microbiota that impacts its adult midgut post-eclosion development, and this is mediated by the insect hormonal circuitry, leading to increased fitness.

## Results

### The midgut of female mosquitoes grows after mating

Female mosquitoes are known to undergo several post-mating changes in behavior and in their reproductive systems[45]. To investigate potential intestinal changes influenced by mating, we initially measured the midgut diameters of mated and virgin *Ae. aegypti* adult females. For consistency, we employed the anterior-to-posterior junction as our reference point and assessed the length at the third circular muscle bundle after this junction (Fig. 1a). We repeated these measurements every 24 h for 9 days to determine if a change in organ size occurred in both mated and virgin females. Adult female midguts were significantly wider in the mated group (mean ± SD = 205.1 ± 30.66 µm), which represents approximately 32% above the average diameter in virgin females (159.55 ± 22.6 µm) (Fig. 1a, b). Interestingly, the treatment of virgin females with JH alone also promoted midgut growth to a size significantly different from that of virgin females (Fig. 1c).

In light of the observed changes in organ size, we looked for possible mechanisms involved in it, such as cell proliferation. Change in size could be related to an increase in cell number supported by intestinal stem cell division. To explore this possibility, we quantified the number of mitotic cells by labeling anti-phospho histone H3 positive cells (PH3 + ). Using our immuno-staining protocol, we counted PH3+ cells in the midgut epithelium for a week after emerging. Unfortunately, on the first day after emergence, viable samples were unattainable due to the fragility of the tissue. Subsequent days' cell counts were insufficient to explain the observed changes in organ size (Fig. 2a). However, recent research with a modified version of the fixation protocol has reported high numbers of PH3+ cells in the midgut of adult females within the initial 24 h post-emergence. This suggests that cell proliferation may indeed be associated with the organ growth we observed. Additionally, this study sheds light on changes in ploidy in midgut cells during the maturation period. Higher polyploid states are generated during this time, potentially contributing to organ growth[42]. This aligns with our findings, as mated sugar-fed mosquitoes showed an average of 13.8 mitotic cells per posterior midgut four days post-emergence, a time when sexual maturity is usually reached (Fig. 2a).

However, virgin females kept a lower number of mitotic cells, averaging close to one cell per posterior midgut and we did not observe any significant increase in PH3+ cells in the first seven days of adult life. Similar to what was found regarding size increase in Fig. 1, JH treatment was able to recapitulate the effect of mating, leading to an increase in PH3+ cells (Fig. 2b). Analysis of transcripts coding for Delta protein, an Intestinal Stem Cell (ISC) cell marker[46] and a key component of the Notch pathway, centrally involved in ISC proliferation and differentiation[47,48], revealed a trend to increase

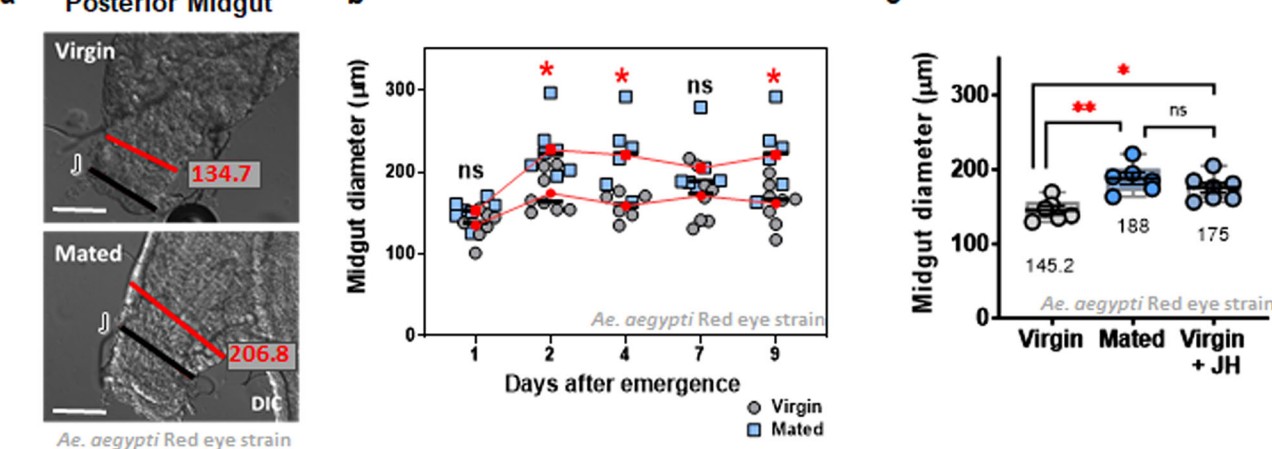

**Fig. 1 | Mating and JH increase midgut size in *Ae. aegypti* adult females.**
**a** Representative images of the anterior to posterior midgut junction (J) on virgin and mated adult females, where the third circular muscle bundle after the junction was selected for midgut size measurement. Images are of differential interference contrast (DIC), and red bars indicate the measurement lengths. Scale bar = 100 µm.
**b** The midgut size (diameter, µm) changes significantly in the first 48 h after emergence in mated females, whereas virgin female midgut size did not see significant changes over time (ANOVA with Turkey's multiple comparison tests). Dots represent individual values, and red points indicate the mean of midgut diameter per group. *P < 0.05 (T-test). **c** Five-day-old virgin females that were treated with JH 1 day after emergence presented midgut sizes comparable to those of mated females. Dots represent the mean value of biological replicates (mean values are written for each group). Black bars present mean and SEM. Boxes in gray present the points of minimal and maximal values. **P < 0.01, *P < 0.05 (T-test). Each experiment was performed at least three times with groups of at least eight individuals per group.

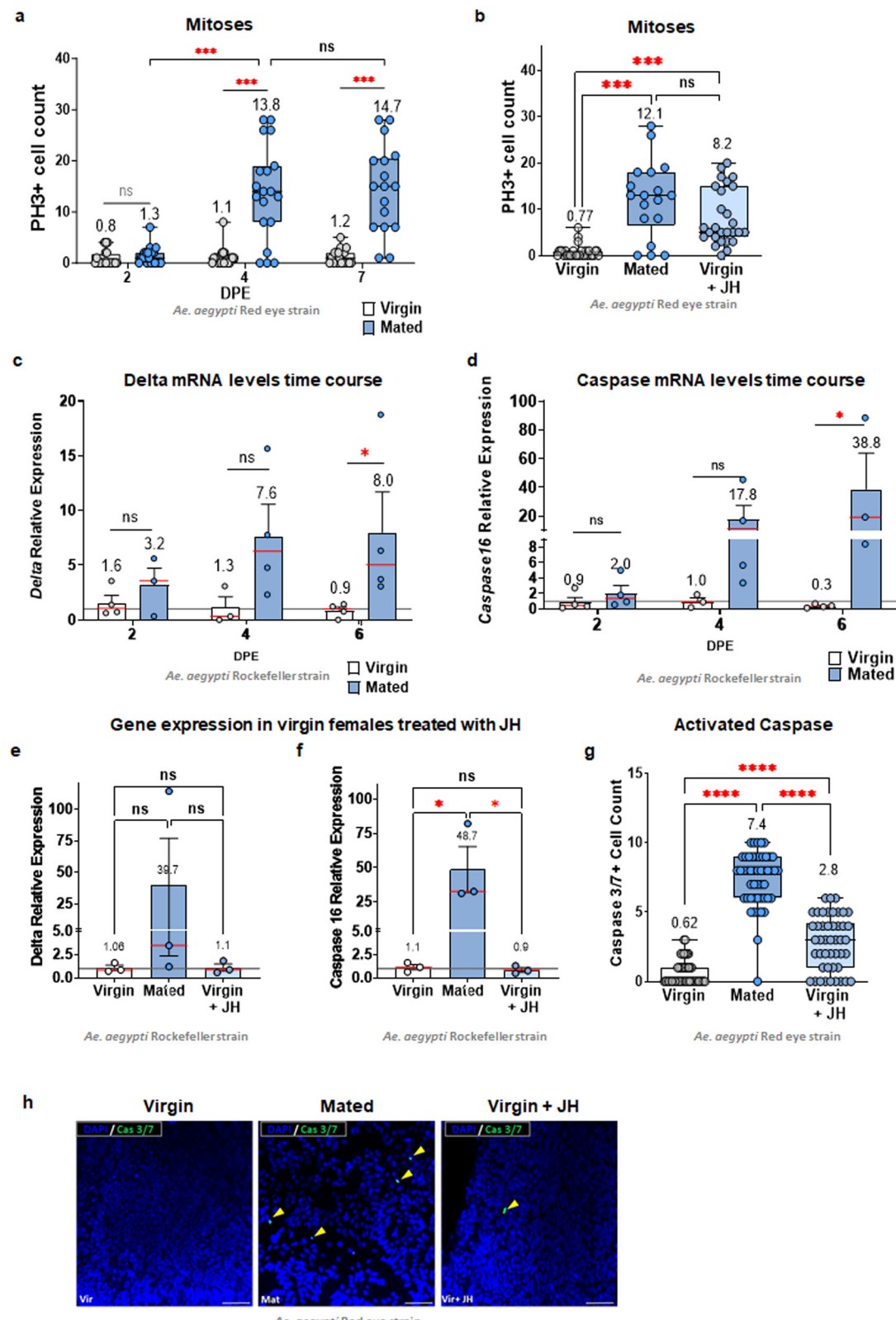

*Ae. aegypti* Red eye strain

expression during the first four days in mated compared to virgin females, but clearly significant differences were attained only by one week after emergence (Fig. 2c).

When looking at the gene expression of caspase-16, a marker of apoptosis signaling in the mosquito gut[49], we observed a trend synchronous to the one for mitosis (Fig. 2d). Interestingly, gene expression in the virgin

females treated with JH was no different from the one found in the non-treated group (Fig. 2e, f). However, detection of the activation of caspases 3/7 to further detail the possibility of apoptotic involvement of the caspases in the gut showed that both mated and virgin treated with JH females had higher cell counts than their virgin counterparts (Fig. 2g, h). This suggests that programmed cell death (PCD) was possibly accompanied by ISC

**Fig. 2 | Mating and JH increase cell turnover in the midgut of *Ae. aegypti* females. a** PH3-positive cell counts between mated and virgin females become different after four days post-emergence (DPE). After this point, the difference between groups was found to be persistent. ***$P < 0.001$ (*T*-test). **b** Measurements of PH3-positive cells, 5 days after emergence and 2 days post-JH treatment, revealed that the gut of JH-treated females presented similar counts as the mated individuals. ***$P < 0.001$ (*T*-test). Each experiment was performed three times with groups of at least eight individuals per group. To estimate ISC overall activity over time, the change in expression of the *Delta* gene by qRT-PCR was evaluated using the value of the virgin females at the first time-point post-emergence as reference (**c**). *$P < 0.05$ (*T*-test). Additionally, to estimate possible cell death, *Caspase 16* expression was also performed using the same parameters (**d**). Additionally, we measured both *Delta* and *Caspase 16* gene expression at day five post-eclosion in JH-treated females, two days post-JH treatment (**e–f**). *$P < 0.05$ (*T*-test), numbers at the top are mean values. In **c–f** panels, bars show SEM, red bars indicate group mean, and dots present mean values per biological replicate. Measurements of Caspase 3/7 positive cells, five days after emergence and two days post-JH treatment (**g**), revealed that the gut of mated and JH-treated females presented higher counts than the virgin individuals. ****$P < 0.0001$ (*T*-test). Each experiment was performed three times with groups of at least eight individuals per group. Representative images of the midgut tissue with Caspase 3/7 staining are shown (**h**). DNA stainings were made with DAPI shown in blue, and Caspase 3/7 shown in green. Scale bar = 10 μm.

mobilization to maintain tissue homeostasis and that JH is not directly involved in the gene expression of the caspases. Rather, it may play a role in modulating other factors contributing to the observed increase in cell counts. Altogether, these observations suggest that the mitotic cell count in the midgut of the adult females is related to a cell turnover mechanism that occurs in mated individuals, which can also be promoted in virgin females treated with JH. Given the average length of the mosquito's adult life, any changes posterior to this period could be caused by senescence and, therefore, were not further evaluated.

### Influence of mating on gut microbiota establishment

Amongst reported changes induced by mating in the gut of *D. melanogaster*, transcriptional regulation of immune-related genes has been observed[50]. In this case, we looked for changes in anti-microbial peptides (AMPs) gene expression to determine if mating and JH influence immunity. Significant reductions in the transcript levels of cecropin-G, defensin, and gambicin were detected in the mated and JH-treated groups (Fig. 3a). The downregulation of AMPs resulted in a significant increase of midgut microbiota in mated and JH-treated individuals, as quantification of 16 S allowed to observe (Fig. 3b, c). No significant differences in ROS levels were detected through DHE assays among the three groups evaluated (mated, virgin, and virgin treated with JH), suggesting that ROS production is not a major antimicrobial mechanism operating in these conditions (Supplementary Fig. 1a). Mated females fed on sugar water containing antibiotics failed to show any significant increase in the number of PH3+ positive cells (Supplementary Fig. 1b), strongly suggesting that the mitotic cells observed were a response to the microbiota expansion in the lumen. In virgin females, the treatment with antibiotics did not affect the number of mitotic cells in the posterior midgut (Supplementary Fig. 1c). As controls, 16 S qRT-PCR to measure bacteria depletion with the antibiotic treatment (Supplementary Fig 2a), and a viability assay (Supplementary Fig 2b) were performed on the groups fed on antibiotics, to verify the absence of microbiota and that the lack of mitotic activity was not due to PCD related to the antibiotics. Interestingly, midguts from virgin females treated with JH also contained bacterial levels comparable to those from mated individuals, suggesting a correlation between JH and microbiota expansion (Fig. 3c). Finally, to explore if the downregulation of these AMPs in the gut impacted the mosquitoes' ability to survive an oral infection, we challenged adult females with *Pseudomonas entomophila*, a known entomopathogenic bacteria. In our experiments, the mated group presented an increased susceptibility to oral infection, showing increased mortality when compared to the virgin groups. Interestingly, the virgin group treated with JH did not show a significant difference from the untreated group in this case, possibly suggesting that other elements from the seminal fluid and the overall mating response are necessary to obtain a significant immune suppression in the gut during an oral infection challenge (Fig. 3d). Together, these results suggest that there is a mechanism in which immunity is modulated to allow a robust establishment of midgut microbiota, likely involving the JH that is passed from males to females during copulation.

### Successful microbiota establishment in the mosquito gut positively influences reproductive output

As mating increases oviposition in blood-fed individuals by modulating gene expression in the ovaries and fat-body[10,14,16,20,45,51,52], our results linking mating and JH to microbiota in the midgut led us to investigate how the microbiota establishment could influence fecundity. To this end, we artificially depleted the gut microbiota by feeding the mosquitoes sucrose supplemented with an antibiotic cocktail (20 units/ml of penicillin, 20 μg/ml of streptomycin, and 15 μg/ml of gentamicin). First, we determined if all three groups would blood feed or not at similar rates in order to assess if mosquitoes had a similar biomass to support egg production. For this, we blood-fed virgin (untreated and treated with JH) and mated mosquitoes and weighed them immediately after engorging. While virgin mosquitoes ate approximately 123% of their own body weight in blood, both mated and virgin treated with JH mosquitoes ate ~115% of their body weights (Fig. 4a). However, there was no significant difference between the net weights of blood consumed amongst all three groups, 3.3 ± 1.1 mg of blood per mosquito. With this in consideration, we blood-fed mosquitoes from the three groups with and without antibiotics to quantify the number of eggs laid per female. Our results show that JH alone is sufficient to induce oviposition in virgins (Fig. 4b), although at a level still lower than control, mated females. However, JH alone was not sufficient to stimulate oviposition in the absence of midgut microbiota. In the mated group, the antibiotic treatment caused a significant decrease in the number of eggs laid per female, but the total numbers were still higher than in the JH-treated group fed with antibiotics. Furthermore, to confirm if the potential quality of the eggs was different, we measured the size (Fig. 4c) and looked at their protein profile (Supplementary Fig 3). Our results show that there were no significant differences amongst egg sizes and electrophoretic profile of the eggs was consistent amongst virgin groups and different from the one from mated females. Total protein content per egg however showed no differences between groups (virgin: 71 ± 1.9 ng/egg, virgin + JH: 68 ± 3.4 ng/egg, and mated: 74 ± 1.5 ng/egg). Finally, we also monitored the longevity of the different groups to determine the long-term effect on fitness that the mating and midgut microbiota establishment had in *Ae. Aegypti* (Fig. 4d). Our results indicate not only that JH-treated and mated female mosquitoes, as previously described by ref. 53, lived longer than virgin mosquitoes, but also that this effect could possibly be related to the presence of the midgut microbiota in these groups (Fig. 4e–g).

In summary, our findings demonstrate that both mating and JH significantly influence the midgut size of *Ae. aegypti* mosquitoes and suggest an immune modulation at least partially guided by JH in this tissue. This immunomodulation in the gut has a substantial impact on mosquito fecundity, as mosquitoes permitting a microbiota expansion in their guts can produce larger quantities of eggs from the same blood volume compared to those without such expansion (Fig. 4h). How JH modulates the establishment of the microbiota in the mosquito midgut to support the demands of reproduction in *Ae. aegypti* is an important part of insect physiology that deserves further attention.

## Discussion

*Aedes* mosquitoes are important vectors of several arboviruses, and understanding the physiology related to blood-feeding and reproduction is key to designing appropriate vector-control strategies. In this study, we address an important knowledge gap regarding the midgut maturation process in *Ae. aegypti* prior to the blood meal. Previous work in *An. gambiae* and *Ae. aegypti* has established a direct regulation between mating and post-

**Fig. 3 | Microbiota of *Ae. aegypti* females is established in the midgut of mated and JH-treated individuals. a** Antimicrobial peptide expression was evaluated in the posterior midgut to determine possible differences in immune modulation amongst the groups. Relative expression was evaluated using the value of the virgin females as reference. **b** Quantification of microbiota by 16 S qPCR on virgin and mated females at different time points revealed a significant difference between the virgin and mated groups. A significant increase of bacteria was observed on mated females. **c** Quantification of microbiota by 16 S qPCR on virgin, mated, and JH-treated females revealed that JH-treated females presented similar bacterial loads to the mated individuals. Each experiment was performed three times, with three samples of at least ten posterior midguts, qPCR data points represent one replicate. ****P* < 0.001 (*T*-test), error bars show SEM. **d** All groups were presented with *P. entomophila* in the food source to challenge mosquitoes to an oral infection. After this challenge, the mated group presented a significant reduction in survival, suggesting that the immune suppression previously observed as downregulation of some AMPs could compromise the mosquito's ability to overcome infection. The experiment was performed in triplicate, with at least 10 mosquitoes per group each time. Data was analyzed with the Log-rank test.

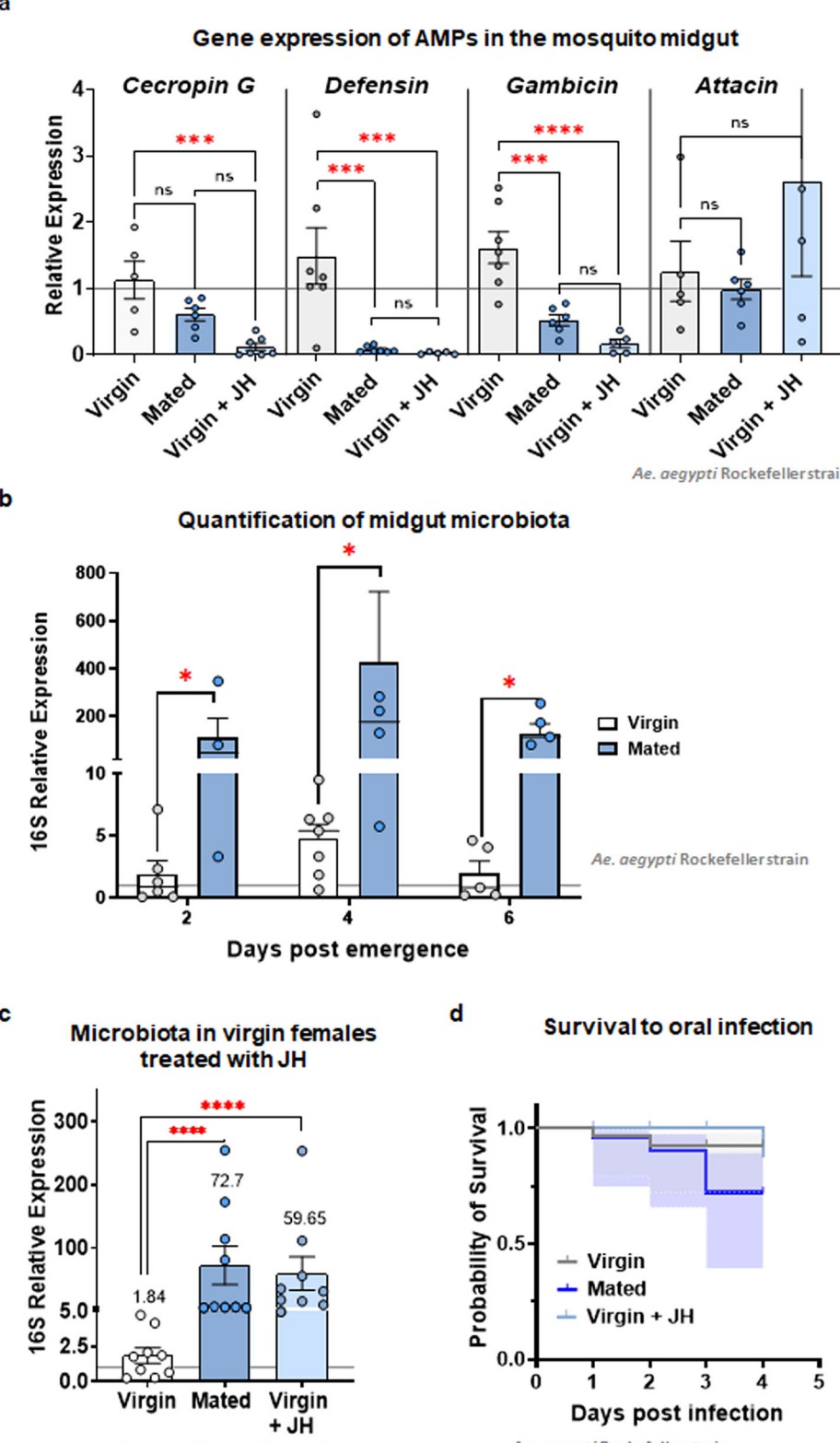

eclosion maturation of the fat body and ovaries to positively influence the reproductive outcome in mosquitoes[10,14,16,17]. However, midgut maturation in response to mating has mostly been described in model organisms, such as Drosophila. In 2015, Reiff et al. described, for the first time, an endocrine remodeling of the female midgut in *D. melanogaster* as a means to support the energy cost of the reproductive demands. The study identified JH as an anticipatory endocrine signal released after mating that acts directly in the intestinal progenitors to obtain a larger organ for nutrient absorption and adjusted gene expression to increase lipid metabolism and improve reproductive output[25]. In mosquitoes, the study of physiological changes after mating has been focused on reproductive tissues and behavior. Here, we report for the first time how the *Aedes aegypti* mosquito midgut can change

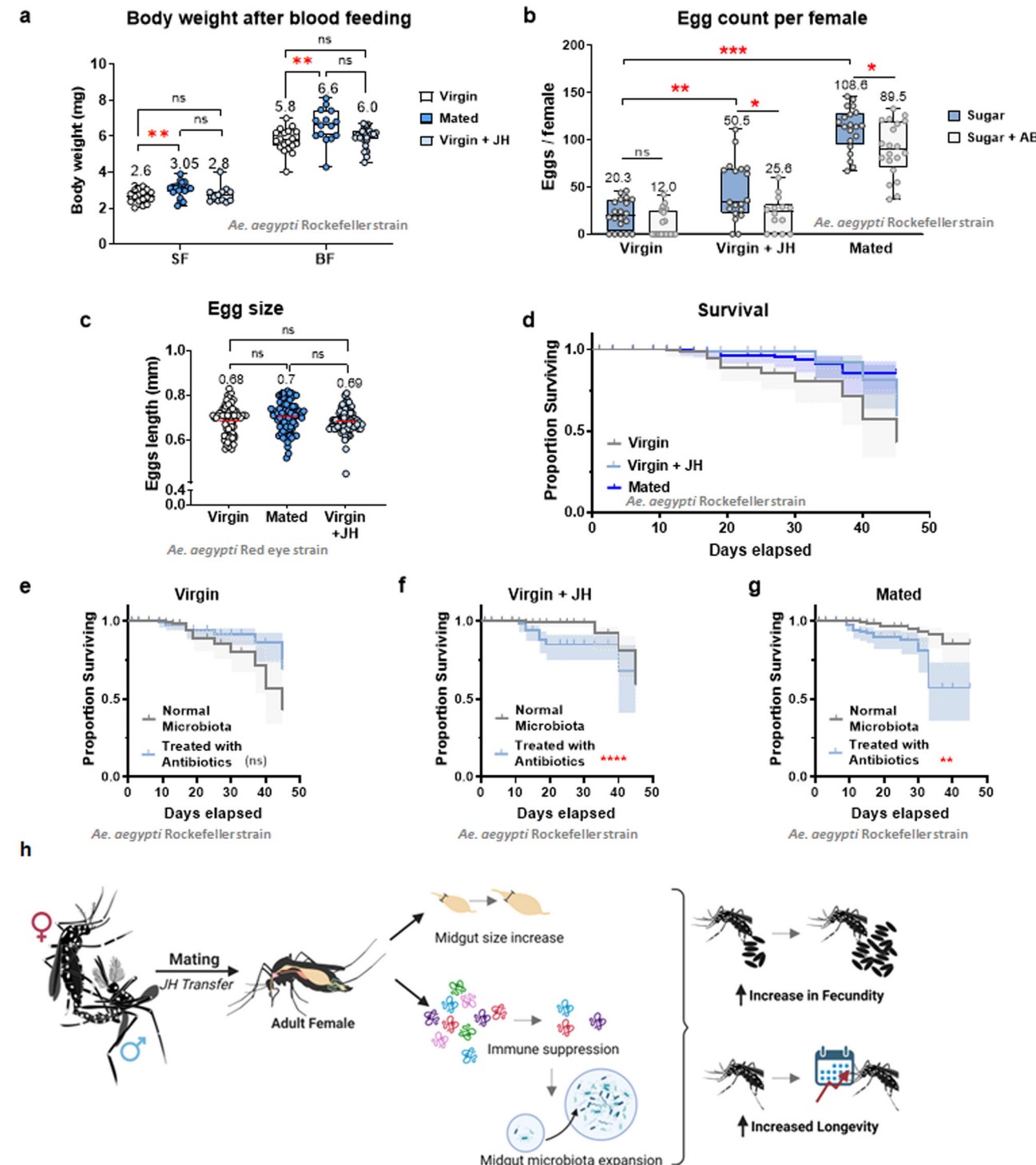

**Fig. 4 | *Ae. aegypti* female fecundity and longevity are impacted by the midgut microbiota establishment induced by mating. a** Mosquito whole body weights were measured in sugar-fed and blood-fed conditions to determine the mosquito's capacity for blood intake while virgin, after mating and treated with JH. A significant difference between sugar-fed and blood-fed mosquitoes was found between the virgin and mated mosquitoes. ANOVA with Turkey's multiple comparison tests, **$P < 0.01$. **b** The number of eggs oviposited by adult females after the blood meal is significantly higher in virgin females treated with JH and mated individuals. However, in both cases, the number of eggs was significantly lower in the group where the midgut microbiota was not allowed to grow. ANOVA with Turkey's multiple comparison tests. **c** Measurement of egg size revealed no differences in length on the eggs laid by virgin, mated, and virgin females treated with JH. ANOVA with Turkey's multiple comparison tests, Ns = $P > 0.05$ (**d**) Survivorship of virgin, JH treated and mated *Ae. aegypti* females. Longrank test for trend, $X^2 = 10.98$; $p = 0.0009$. The shaded area represents a CI of 95%. $n = 40$; two replicates. **e–g** Survivorship of virgin, JH treated and mated *Ae. aegypti* females when the midgut microbiota establishment is impaired by antibiotic treatment [grey]. Longrank test for trend ($X^2 = 0.4985$; $p = 0.480^{2)}$), ($X^2 = 16.63$; $p < 0.0001$), ($X^2 = 13.88$; $p < 0.0002$) respectively. The shaded area represents a CI of 95%. $n = 40$; two replicates. In summary, mating and JH treatment alone cannot only induce an increase in midgut size in *Ae. aegypti* (**h**), but also cause immune suppression in the form of reduced expression of some AMPs that regulate the establishment of midgut microbiota. With these effects, the mated females show an increase in fecundity and longevity when compared to virgin individuals. Image created with BioRender.com.

in response to mating. The midgut diameter of copulated *Ae. aegypti* adult females increases, potentially increasing the size of the blood meal and the number of eggs produced[54], therefore increasing fitness.

In our experiments, the increase of fecundity in *Ae. aegypti* females could also be induced by JH treatments, providing additional evidence of the important role that JH plays in this mosquito in the post-mating response. In 2014, Clifton et al. showed that male mosquitoes can alter females' resource allocation priorities towards reproduction by transferring JH during copulation[9]. Other previous studies have also shown post-mating changes in reproductive organs by JH or seminal fluid delivery[15,19,45]. Here, we show that JH-mediated signaling can elicit substantial changes in the midgut of *Aedes aegypti* to support reproduction. The implications of mosquitoes having larger blood meals in the context of disease transmission could be significant, as with a larger volume of blood, the probability for infection could increase, and this could deserve further attention.

The last 20 years of research have changed the old view on the "commensal microbiota" found in the digestive apparatus just as a mere cohabitant using the leftovers from metazoan food. Previously unappreciated complex microbial communities have been found ascribing to the microbiome, central roles in the biology of the multicellular host, essentially as a component of the physiology of several organs, but especially of the digestive apparatus[55]. Here, in addition to the effect of mating on organ size, we also detected a significant increase in bacteria abundance on the midgut of mated females, revealing a cross-talk between hormonal control of insect development and the microbial community. We sought to obtain a global measurement of total bacteria present on the mosquito midguts by quantifying 16 S and obtained relative expressions by normalizing with housekeeping genes from mosquito tissue. This strategy stems from the assumption that if the epithelium surface increases, it would maintain proportionality in the comparison. Under this assumption, our results show that there is an increase in bacteria abundance per midgut in the mated and virgin treated with JH females; however, it is less clear if the proportionality remains when comparing microbiota per midgut volume.

In the trade-off between immunity and reproduction, changing resource allocation is a central idea in the theoretical framework of evolution of immune response and host-pathogen (symbiont) interaction. In mosquitoes, ecdysone and JH signaling in the fat body after a blood meal inhibit innate immunity response to direct resources for reproduction[22,56]. While our study highlights the immunosuppressive effects of JH signaling in the midgut, it is crucial to consider that the interplay between mating-derived factors and the mosquito's immune system is likely multifaceted. Beyond JH, the seminal fluid in mosquitoes contains a myriad of components that could potentially influence the responses[47]. Regarding vector competence, in *Anopheles coluzzi* mosquitoes, mating increased the susceptibility to Plasmodium falciparum infection, which could also be induced by the steroid hormone 20-hydroxyecdysone (20E) alone[57]. JH synthesis in mated *Drosophila* females results in compromised immunity, as mated females are more likely to die from infection and present higher pathogen loads[58]. Additionally, Ahmed and collaborators reported in 2020 a steroid signaling pathway from the ovaries to the gut to promote growth, especially when *Drosophila* females are mated[23]. Here, we show that mating and specifically JH signaling down-regulates expression of AMPs (cecropin G, defensin, and gambicin) in the midgut, and this immunosuppressive (tolerogenic) regulatory loop is critical to allow pre-blood meal expansion of the indigenous microbiota. Changes in the expression of attacin were not observed, and transcriptomic analysis of this gene suggests it may not have a significant role in the immune modulation of the gut epithelium, as it is mostly expressed in other tissues[59]. Moreover, the AMPs expressed in the gut epithelium seem to control the microbial loads, and even the smaller amount of microbiota of the virgin female appears to impact mosquito fitness, as antibiotic treatment of virgin females tends to increase lifespan. This occurs in contrast to either JH-treated or mated females that display a decrease in lifespan upon antibiotic administration. This result indicates that the building of the mutualistic feature of the symbiosis between the mosquito host and its microbiota is developmentally regulated by mating and mating-derived hormonal signaling.

Our results confirm that the midgut epithelium of *Ae. aegypti* responds to post-mating signaling and that this response is involved in the overall increase in reproductive output. The process involved in this complex phenomenon concerns not only endocrine signalization triggered by JH transferred during mating but also the microbiota resident in the midgut lumen. In the context of these complex interactions, it is also noteworthy to consider the effects of the tissue turnover observed and the potential role of programmed cell death (PCD). Although PCD is conventionally associated with tissue size reduction, our results suggest that the level of PCD detected in the midgut may be a secondary outcome to the microbiota in the lumen, and our time course measurements suggest it is not sufficient to impact overall organ size in the conditions tested. Furthermore, the possibility of a non-apoptotic role of the caspase activation detected, as it has been described in some other tissues with the function of preserving tissue homeostasis[60,61], has not been explored and could be possible.

An interesting question raising from our findings is whether the establishment of the gut microbiota in the mosquito gut could also play an adaptative role to improve the outcome of each blood meal and whether other autogenous mosquitoes use the same strategy. Furthermore, in this study, we did not evaluate the composition of the gut microbiota in the virgin, virgin treated with JH or mated individuals, but further characterizations of these could be useful to determine if particular bacterial species are favored by the post-mating endocrine modulation to the gut to improve blood meal utilization and increase the reproductive output. Recognizing that laboratory-reared mosquitoes are also exposed to very limited microbiota diversity, performing such studies with field mosquitoes and field water, and microbiota samples are the next steps to fully characterize how mosquitoes modulate their midgut microbiota to potentiate fecundity in their normal habitats.

Transcriptomic studies by region of the gut and using single-cell analysis to determine if a particular section of the gut or a particular cell type responds to the mating signalization and how they do it will also be extremely important to understand the mechanism. Characterization of the cellular profiles in both virgin and mated individuals to determine if percentages of the midgut epithelium-specific cell types change after mating is also needed. With this, determinations of the infection rates and vector competence of the mosquitoes will help design comprehensive vector control strategies.

## Methods

### Ethics statement

At the Federal University of Rio de Janeiro (UFRJ), all animal care and experimental protocols were conducted in accordance with the guidelines of the Committee for Evaluation of Animal Use for Research (CAUAP-UFRJ). The protocols were approved by CAUAP-UFRJ in 2013 under registration number IBQM155/13. At the Centers for Disease Control and Prevention (CDC), laboratory stocks and rearing protocols for the *Ae. aegypti* used were obtained from the Biodefense and Emerging Infections Resources/ Malaria Research and Reference Reagent Resource Center (BEI/MR4). We have complied with all relevant ethical regulations for animal use.

### Mosquito rearing

For this study, two *Ae. aegypti* strains were used. Red-eye strain, at the Federal University of Rio de Janeiro, Rio de Janeiro (UFRJ), Brazil, with known Dengue virus susceptibility. Rockefeller strain, also a Dengue virus-susceptible line, was used at the Centers for Disease Control and Prevention (CDC), Atlanta, Georgia, USA, and at Cornell University, Ithaca, New York, USA. Red-eye strain was reared at the UFRJ, under a 12:12 light-dark cycle, at 27–28 °C and 70–80% relative humidity. *Ae. aegypti* Rockefeller strain was reared at the MRA-112 BioDefense Emerging Infections (BEI), Malaria Research and Reference Reagent Resource Center (MR4) and at Cornell University under a 12:12 light-dark cycle with a 30-minute dawn and dusk

period at 27 °C, 80% relative humidity. In all cases, adults were fed with 10% sucrose *ad libitum*. Gut size measurements, counts of mitotic and caspase 3/7 positive cells, quantification of ROS levels, and egg protein content measurements were performed at UFRJ using the Red-eye strain. Gene expression analysis and fecundity and longevity assays were performed at CDC using the Rockefeller strain. Caspase 3/7 positive quantifications, blood-feeding weight measurements, gene expression analysis, and bacterial infections were performed at Cornell University using the Rockefeller strain. In all cases, pupae were manually sexed, and female pupae were isolated prior to emergence for any experiments requiring virgin females. Blood-feeding was performed through Parafilm membranes using defibrinated sheep blood in water-jacketed artificial feeders maintained at 37 °C. The insects were starved for 4 h prior to feeding. Spermatheca of each individual mosquito was dissected and collected to be examined under a 40X objective in a microscope (ocular magnification of 400X) to validate the mating status for all experiments requiring mated females, and all individuals without visible sperm were discarded from the mated groups.

## JH and antibiotic treatments
JH treatments were performed by topically adding 0.5 µl of 1.0 µg/µl solution of JH III (Sigma-Aldrich, MO, USA) in 100% acetone (Santa Cruz Biotechnology, TX, USA)[62] to the VII abdominal segment of cold-anesthetized 1-day-old mosquitoes. Virgin females of the same age and cohort were used as control and were treated with acetone alone (in all figures, this group is labeled as "Virgin"). 10–15 mosquitoes per group were used in at least three separate biological replicates. Dissections were made 24 h after JH treatments. For antibiotic-feeding experiments, mosquitoes were rendered free of cultivable bacteria by maintaining them on a 10% sucrose solution with 20 units/ml of penicillin, 20 µg/ml of streptomycin, and 15 µg/ml of gentamicin as previously described by ref. 63, beginning from the first-day post-emergence until the time of dissection. Verification of the elimination of free cultivable bacteria was also performed as described by these authors. Briefly, 5 midguts from each group were dissected and homogenized in 500 µl of PBS, and serial dilutions were plated on LB agar to confirm bacterial elimination. Groups were composed of 15 mosquitoes, and three biological replicates were performed. All cages used in these experiments were cleaned with 70% ethanol prior to use, and sugar pads were changed daily.

## RNA extraction and qPCR analysis
Posterior midguts from *Ae. aegypti* adult females were dissected on sterile ice-cold phosphate-buffered saline (PBS) pH 7.4, at different time-points after emergence (2, 4, 6–10 days, depending on the experiment). For every sample, RNA was extracted from three pools of ten posterior midguts (*Ae. aegypti* red-eye strain), in at least three biological replicates, using TRIzol (Invitrogen, Carlsbad, CA, USA) according to the manufacturer's protocol. RNA from three pools of ten posterior midguts (*Ae. aegypti* Rockefeller strain), and at least three biological replicates, was extracted using RNeasy Micro kit (Qiagen, Germantown, MD, USA), according to the manufacturer's protocol. In both cases, complementary DNA was synthesized starting with 1 µg of RNA using the High-Capacity cDNA Reverse transcription kit (Applied Biosystems, Foster City, California, USA). The qPCR was performed with the StepOnePlus Real-Time PCR System (Applied Biosystems) or a QuantStudio6 Real-Time PCR System (Applied Biosystems), using the Power SYBR-green PCR master MIX (Applied Biosystems). The *Ae. aegypti* ribosomal S7 gene and actin were used as endogenous controls. All oligonucleotide sequences used in qPCR assays for target genes were synthesized by Integrated DNA Technologies, Coralville, IA, USA, and are available in Supplementary Table 1. Particularly, for all experiments involving quantification of 16S[64] by qPCR, all the dissection equipment was autoclaved before use (forceps, tubes, PBS, etc.) and dissections were carried out inside a Class II biological cabinet, to reduce environmental contamination. Additionally, RNA extraction, reaction preparation of cDNA synthesis and PCRs are also performed under the same conditions. The Comparative Ct Method[65] was used to compare the changes in the gene

expression levels. As the basal or initial state of all groups of adult females is the virgin condition, this group was the one used as control, and therefore, all horizontal lines at y = 1 correspond to the average expression in this group. In experiments where time courses were performed, the virgin group at the earliest timepoint was used as reference.

## Midgut size, Intestinal Stem Cell (ISC) proliferation and apoptosis
Groups of at least eight full midguts (anterior and posterior) per condition were dissected on ice-cold PBS at different time points after the emergence, fixed, and treated for immunostaining as previously described[34]. For size measurement, differential interference contrast (DIC) images were acquired with a Zeiss AxioObserver (Zeiss, Oberkochen, Germany) under a 20X objective. A single field was acquired at the anterior/posterior midgut junction, using the measuring tool from the Axio Vision software (Zeiss) we determined the diameter of the posterior midgut at the third circular muscle bundle after the junction (posterior muscle organization was defined as described by Park and Shahabuddin, 2000)[66]. In *Drosophila*, the diameter of the midgut part adjacent to the hindgut (which could be called the "posterior") had been used previously to infer the size of the organ by ref. 25. In mosquitoes, the midgut has a very clear division of anterior and posterior, with functional differences between these. The posterior midgut expands to accommodate an incredibly large amount of blood, and therefore, we considered it important to obtain measurements of this organ at the critical point for food storage. The shape of the mosquito posterior midgut, which resembles a larger sac rather than a tube as the anterior midgut, is harder to mount without any wrinkles or twists that could introduce noise in the length measurements. Therefore, measuring the midgut diameter at the anterior to posterior junction is the best proxi we could find to measure organ size.

Quantification of mitosis in the midgut of adult females (red-eye strain) was performed using primary antibody mouse PH3 (Cat No. 05-598; 1:500, Merck Millipore, Darmstadt, Germany) as previously described[34], with a Zeiss Axioskop with an Axiocam MRC5 using a Zeiss-15 filter set (excitation BP 546/12; beam splitter FT 580; emission LP 590) and 20X objective. For the acquisition of the gut images, a single field was acquired at the anterior to posterior midgut boundary, using Axio Vision software (Zeiss). To visualize apoptotic cells using activated caspase 3/7, we incubated mosquito midguts with 5 uM of CellEvent™ Caspase-3/7 Green Detection Reagent (Cat. No. C10427; Thermo Fisher Scientific) according to the manufacturer's instructions. Cell nuclei were counterstained with 4′,6-diamidino-2-phenylindole (DAPI) (1 µg/ml) (Sigma, MO, USA), and guts were mounted for confocal microscopy in VectraShield anti-fade medium. The midguts were observed on a Zeiss Elyra SR-SIM microscope (Zeiss, Germany) with a 100x oil immersion objective. Images were acquired with ZEN software (Zeiss) with a SIM analysis module. Images were generated using Maximum intensity projection and representative images shown contain 12 optical sections and a scale bar of 50uM.

## Reactive Oxygen Species (ROS) production by midgut tissue
To assess ROS levels in mosquito midguts, at least ten midguts of virgin, JH-treated, or mated *Ae. aegypti* Red-eye strain adult female groups were incubated with a 2 mM solution of the oxidant-sensitive fluorophore DHE (dihydroethidium) (Invitrogen) diluted in Leibovitz-15 media supplemented with 5% fetal bovine serum for 20 min at room temperature in the dark, at room temperature. After incubation, the midguts were washed in a dye-free medium and transferred to a glass slide in a drop of PBS for epifluorescence or confocal microscopic examination. Midguts were examined with a Zeiss Axiovision 40 with an Axiocam MRC5 using a Zeiss-15 filter set (excitation BP 546/12; beam splitter FT 580; emission LP 590). Differential interference contrast (DIC) images were acquired with a Zeiss AxioObserver. In three biological replicates, a comparison of fluorescence levels among distinct images was performed under identical conditions (objectives 20X and exposure times) using the same objectives, microscopes, and similar exposure times in each experiment.

## Cell viability assay

Pools of 3–5 posterior midguts from sugar-fed or sugar and antibiotic-fed *Ae. aegypti* Red-eye strain adult females were dissected in PBS and quickly placed on 100 μL of L15 media supplemented with 5% of fetal bovine serum (FBS) were incubated in a 96 well plate for up to 2 h. Three biological replicates with at least a technical triplicate were used for all conditions. Immediately after this incubation, the media was removed and replaced with 100 μL of 0.5 mg/mL MTT (3-(4,5-Dimethylthiazol-2-yl)-2,5-Diphenyltetrazolium Bromide) solution (Sigma–Aldrich) in DMEM (Dulbecco's modified Eagle's medium) and PBS buffer, (9:1 v/v). All samples were incubated at 37 °C, protected from light for 30 min. and formazan crystals were solubilized with 100 μL of 100% DMSO (Dimethyl sulfoxide), mixing gently for 15 min. Optical density was quantified spectrophotometrically at 540 nm on a Spectra Fluor plate reader (Spectra Fluor, Tecan, Austria). Cell viability was calculated based on the measured absorbance relative to the absorbance of tissues incubated directly on the MTT solution, which represented 100% cell viability, and with midguts dissected, immediately frozen at −70 °C and boiled for 5 min, as a negative control.

## Blood feeding capacity assay

Female pupae, Rockefeller strain, were separated into the following groups: virgin and virgin with JH (10–15 per biological replica). Additionally, non-sexed groups of 25 pupae (containing approx. 12 females each) were also used. Groups with JH were treated as previously described, and all groups were fed on 10% sucrose solution and kept at the previously described rearing, temperature and humidity conditions. Mosquitoes were fed on defibrinated sheep blood (Lampire, PA, USA) in water-jacketed artificial feeders maintained at 37 °C, seven days after emergence. Mosquitoes were allowed to feed for 5 min, and only fully engorged mosquitoes were included in this study. Weight measurements were performed individually as previously described by ref. [16]. Briefly, fed females were transferred to a 15 mL falcon tube and flash-frozen in liquid nitrogen for 30 s to prevent loos of fluid from diuresis. Non-blood-fed individuals for each treatment group were also collected to provide baseline weights. Females were weighed individually on a Cahn C-31 microbalance to determine their mass in mg (Cahn Instruments Inc., Cerritos, CA, USA).

## Oral Infection

*Ae. aegypti* adult females, in groups of virgins, virgins treated with JH and mated, were orally infected with *Pseudomonas entomophila* with an OD600 of 100 as previously described in ref. [42]. Briefly, a cotton ball was soaked in a 1:1 mixture of 20% sucrose and concentrated bacterial culture (OD$_{600}$ 200). 5-day-old mosquitos, that had been starved for at least 12 h, were offered the sugar mix with bacteria, which was removed after four hours. Mosquitos were kept on 10% sucrose prior to infection and following infection. Infected mosquitos were assessed for survival 24, 48, 72, 96, and 120 h after infection. At least three biological replicates were performed with 7–10 mosquitos per group.

## Reproductive output and longevity assays

Female pupae, Rockefeller strain, were separated for the following groups: virgin, virgin with antibiotics, virgin with JH, virgin with antibiotics and JH (20–25 per biological replica). Additionally, non-sexed groups of 50 pupae were (containing approx. 25 females each) were separated to obtain a mated, and a mated with antibiotics groups. Groups with JH were treated as previously described, and all groups with antibiotic treatment were kept on sugar water containing the antibiotic cocktail from the emergence day (day 0) to the moment of blood-feeding. All groups were fed on 10% sucrose solution and kept at the previously described rearing, temperature and humidity conditions. Mosquitoes were fed on defibrinated sheep blood (Hemostat Laboratories, CA, USA) in water-jacketed artificial feeders maintained at 37 °C, seven days after emergence. Fully engorged females were immediately aspirated and placed in individual containers to monitor oviposition. Oviposition cups with DI water were placed on the containers for up to 3 days, 48 h after the blood meal, and each group was maintained on their original sugar-water condition (with or without antibiotics). After the oviposition cycle, females were placed together in the original groups and monitored for longevity for 45 days, recording data every other day, or every third day on weekends. During this time, cotton balls with the respective sugar treatment were changed daily.

## Protein extraction and analysis from fresh laid eggs

*Ae. aegypti* red eye strain adult females (virgin, virgin with JH, and mated) were blood-fed as previously described, and two days later, 80 fresh laid eggs were collected and homogenized in a Potter-Elvehjem tissue grinder in 0.2 ml of a mixture of protease inhibitors (0.05 mg/ml of soybean trypsin inhibitor, 0.05 mg/ml leupeptin, 1.0 mM benzamidine, and 0.01% phenylmethylsulfonyl fluoride (PMSF), purchased from Sigma) in 20 mm Tris-HCl buffer (pH 7.4). Egg homogenates were centrifuged at $15,000 \times g$ for 5 min at 4 °C. The crude egg extract supernatant was used for protein profile analysis. Protein concentrations of egg homogenate were determined by the method of Lowry et al. using bovine serum albumin as standard[67]. 40 μg of protein were subjected to electrophoresis on 15% SDS-PAGE gels with a constant voltage of 90 mV. The gels were stained with Coomassie Blue G and destained with distilled water plus methanol 40%[68].

## Statistics and reproducibility

An unpaired *t*-test was utilized where comparisons were made between two treatments. Where more than two treatments are analyzed together, a one-way ANOVA (analysis of variance) test followed by Tukey's multiple comparison test was utilized. For survival curves, long-rank (Mantel–Cox) test with long-rank test for trends were performed. All statistical analyses were performed using GraphPad Prism Software version 9 (Graphpad Software, CA, USA).

## Reporting summary

Further information on research design is available in the Nature Portfolio Reporting Summary linked to this article.

## Data availability

The data that support the findings of this study are available at https://figshare.com/s/375015cb97d894253cce and from the corresponding author, M.L.T. and G.O.P.S., upon reasonable request. The non-cropped and non-edited file pertaining to the protein gel from Supplementary Fig. 3 is available in Supplementary Fig. 4.

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

## Acknowledgements
We thank all members of the Laboratory of Biochemistry of Hematophagous Arthropods, especially Jaciara Loredo, Mônica Sales and S.R. Cassia for providing technical assistance. We would also like to thank Dr. Fernando G. Noriega for his important contributions to the experimental design and helpful discussions. We thank the staff and scientists within the Entomology Branch and the Division of Parasitic Diseases and Malaria, especially Catherine Steele for her assistance in the data collection and Dr. Priscila Bascuñan for the helpful suggestions and technical assistance. We also thank Dr. Nicolas Buchon for providing the bacterial strain for the mosquito infections. Finally, we thank Dr. Laura Harrington for her technical feedback. All primers used in this work were provided by the Biotech Core facilities at CDC. The findings and conclusions in this report are those of the authors and do not necessarily represent the official position of the Centers for Disease Control and Prevention.

## Author contributions
M.L.T., P.L.O., and G.O.P.S. conceived and designed the experiments. M.L.T., A.B.W.N., V.B.R., A.P.G.M., K.X., and S.S. conducted experiments. M.L.T., E.M.D., and G.O.P.S. analyzed the results, provided funding, and all authors interpreted the results. M.L.T. designed the figures and wrote the manuscript. All authors reviewed the manuscript.

## Competing interests
The authors declare no competing interests.
