## [Peer Review File · Communications Biology]

Reviewers' comments:

Reviewer #1 (Remarks to the Author):

Review Commsbio Taracena et al.

In the manuscript, 'Aedes aegypti midgut microbiota establishment in adult females is regulated by Juvenile Hormone to enhance fecundity and fitness', Mabel Taracena and colleagues describe the interesting finding that in the mosquito intestine Juvenile Hormone suppresses the immune response e.g. Anti-Microbial Peptides. JH is known to be boosted upon mating in fruit flies and Taracena et al. discover that the expansion of epithelial tissue leads to more egg production and life expectation, which is very interesting as the observed downregulation of antimicrobial peptides might also cause opposing effects. Overall, the manuscript is well and comprehensively written and data presentation is clear and sound and includes novel findings. However, I have some major and minor remarks concerning the current version of the manuscript, as it is preliminary/incomplete in its writing in the final paragraphs of e.g. results and one method.

Major:

- Page 12, Line 24-27. Fig.2D The increase in apoptotic signaling is a very interesting finding. Similar observations have been made in the Drosophila midgut, where Notch and EGFR signaling control progenitor apoptosis to balance ISC cell production with actual epithelial cell need (Reiff et al., 2019). As Aedes does not offer the genetic tools for a study of signaling processes with cellular resolution, the authors might have a look whether actual caspase-3 + cells are found and aim to identify whether these cells are morphologically resembling progenitors or enterocytes. It could also briefly be discussed as PCD during mating induced midgut expansion seems counterintuitive.
- Page 12, Line 24-27. Fig.2D It should also be noted that expression of a caspase does not equal its activation. The Baena-Lopez lab also revealed non-apoptotic roles for caspases in midgut homeostasis that should be discussed.
- Page 15+16 the text does not refer to any panel of Fig.4 apart from 4A. This part definitely needs rework. From an experimental side of view, its hard to see in the 4B whether there is differences, which is better in 4C-E.
- Page 15 to 16, Line 6 following is incomplete
- From M&M and the main text, it does not get entirely clear how 16s as proxy for bacterial number for midgut microbiota is calculated, which raises the next question. Is this referred and normalized to the also observed size increase? So actually, more microbiota per midgut? Or per midgut volume?

Minor:

- Page 3, Line 23-25 & Page 19, Line 19-21, actual hemolymph 20HE titer increase upon mating was shown in Zipper et al., 2020.
- For clarity in all figures, in quantifications of JH fed VF, that should be clearly visible for the reader without the text, like in Fig.4A. In other figures (like Fig2) it just states JH which could be misleading.

Reviewer #2 (Remarks to the Author):

The authors showed that mating and JH induce gut growth by stimulating cell proliferation in *Aedes aegypti*. This causes reduction of AMP expression, establishment of gut microbiota population. This finally results in increased reproductive outcome and longer lifespans. Although I agree with the authors' interpretations of the presented data, I see several things are missing to connect all these observations. Please see details below.

Major points

- Although it is clear that JH changes the gut size (maturation) by regulating cell proliferation, in order to show this causes reproductive outcomes, it is key to show this leads to a blood meal size, which is not directly shown in this manuscript.
- The other important thing missing in this manuscript is whether mating results in an increase of JH concentration in circulations. Is it possible to look at *jhamt* and *kr-h1* expression in virgin (0 day), 0.5, 1 and 2-day post mating? Since the gut size becomes larger by 48 hours, these four time points must be sufficient to detect expression of these genes if this happens.
- Although I agree mating leads to reduced AMP expression, which results in increased 16S expression, it is not clear how biologically relevant this reduced AMP expression is. It is because AB treatment is not necessarily the same thing. I would like to see whether this reduced AMP expression increases lethality by oral bacterial infection. This immunity-reproduction relation is a classic trade-off that can fundamentally support the authors' conclusion.
- It is not clear whether unfertilized eggs by virgin mosquitoes that are treated with JH are qualitatively/quantitatively similar to fertilized eggs. Is it possible to see the size and (if possible) egg contents? I am asking because applying JH may simply induce oviposition of immature eggs.

Minor points

- Line 94. Typo? Established?
- Figure 1C, 2B and so on. It is not clear whether the "virgin" means "virgin + acetone" and whether "mated" means "mated = acetone". It is not clearly written besides the methods. If you compare with JH treated guts, this "virgin" has to be "virgin + acetone" and "mated" has to be "mated + acetone". Please clarify.
- Figure 3A, B and C are mixed up in the main text.
- Lines 420-421. Typo. "we" twice.

Reviewer #3 (Remarks to the Author):

The manuscript titled "*Aedes aegypti* midgut microbiota establishment in adult females is regulated by Juvenile Hormone to enhance fecundity and fitness" by Taracena et al. presents effects on the female midgut caused by mating and artificial application of JH (a component of materials transferred from male by mating). The authors examined various aspects of effect of mating and JH utilizing a wide range of techniques. The maturation of the midgut prior to the blood meal has long been overlooked and this study sheds light on the potentially critical part of female *Ae. aegypti* physiology. The results presented in the manuscript are new and of interest of the research community. However, the manuscript needs to

improve with validation of used techniques, a better organization, and ordered presentation to be convincing. The manuscript would attract more interest with discussion of other factors transferred by mating than JH, which seem also play important roles in midgut maturation. A brief mention on the negative results (attacin) could also support the midgut undergoes complex modulation of gene expression during maturation triggered by mating.

In addition, I would appreciate that authors would have been a little more careful in preparing the manuscript as many points I found could have been caught by themselves.

Points:

Title:

It appears that JH is not only the factor for microbiota establishment from the results, and the title may be misleading.

Abstract:

I feel it needs more background (why it's done, what questions asked) as it is mostly results and conclusion.

Line 37: "vector capacity" vectorial capacity?

Introduction:

Line 60-61: To be accurate, mosquitoes do not transmit diseases, but pathogens.

Line 66: "its development" Do mosquitoes need blood for development? Or authors mean they need blood for development of oocytes?

Line 78: "this tissue" which tissue (fat body or ovary)?

Line 94-95: "it is well established that newly emerged mosquitoes do not feed on blood" If so, references needed.

Line 111-112: "As increased fecundity and longevity after mating had already been described in this organism" requires references.

Materials and Methods:

Line 123: "A. aegypti" please be consistent throughout the manuscript: *Ae. aegypti* or *A. aegypti*.

Line 126-132: This study used two different strains of *Ae. aegypti* and the experiments were conducted in two different laboratories. In the methods, results and discussion, no clear description of which experiments were conducted with which strain/lab, or if the same experiments were conducted in both strain/lab. I think that, at least, clear statement of which strain/lab was used for which experiment is needed to avoid future confusion. If they observed differences of results by strains, that may also be of

interest of researchers in the same field. It is ideal to have justification of use of two different strains for the study.

Line 135: “sheep blood” is it the same “defibrinated sheep blood” appears later in the manuscript?

Line 146-149: Please provide reference(s) if this method has been previously established (with verification of free of cultivable bacteria). Otherwise, authors may need to mention that they have verified that there is no cultivable bacteria in the gut (with description of its method).

Results:

Line 250-253: Authors did not mention on the establishment of anterior-posterior midgut junction for the size of the (entire) midgut as it is so mentioned in the legend. Reference 46 does not seem to describe this. If it has been done previously, please cite.

Line 256, 257: are the numbers mean \pm SD?

Fig1A: Red letters are difficult to see. Both panels seem to have another faint red lines, which are confusing.

Fig1B: Lacks the units on x-axis and the significance notation (asterisks etc.).

Fig1C: no description of the numbers shown in the figure (mean, or median?). No explanation of dots, lines, boxes, and single * are given.

Line 262-269: Not sure why this is here. It seems discussion about following results (line 283-289), and sounds odd here.

Line 298-299: seeing the Fig2B, there may also be other factors transferred by mating contributing the phenomenon, and authors did not test Delta expression and caspase 16 for JH-treated virgins.

Fig2: as for Fig1 no explanation of dots, lines, boxes are given. Only ** is explained, though *, ***, and ns are also shown.

Fig2C-D: relative expression to What? What are a horizontal line across the panels at y=1?

Line 322-323: Fig3A is 16S quantity, not AMPs. (looking at Fig3C) significant difference seems to be only between virgin and JH (they seem to lack significance notation between virgin and mated).

Line 323-326: Fig3B is 16S, and Fig3C is AMP.

Supplemental figure 1 legend:

Line 667: what is “d ()”?

SFig1A contains a scale bar, but no description of the size of the bar is given.

Line 336-339: Assuming (the current) Fig3A shows 16S abundance (expression?) relative to host S7 (or actin), the change (expansion) ratio between 2 and 4 or 6 DPE for both virgin and mated groups are similar (thus virgin group's bacteria also expanded at similar rate to mated group *calculation below). So it looks, to me, that the cell proliferation (and effect of JH) may have more correlation to abundance of bacteria, rather than expansion.

*Ratio calculation (from current Fig3A):

4 vs 2 DPE: virgin: $4.8/2.0 = 2.4$; mated: $425.3/112.4 = 3.8$

6 vs 2 DPE: virgin: $2.0/2.0 = 1$; mated: $128.3/112.4 = 1.14$

Fig3: the figures do not match either legends or the text. The reference(s) of relative expression are not clarified. No explanation of bars, dots and significance notations (but ***) are given. What is the horizontal line across the panels at $y=1$?

Fig3: No discussion about attacin are found in the results or discussion.

Fig4A: no explanation of dots, box with whisker, and significance notation is given in legend.

Fig4B: missing significance notation (between which are/is significantly different at what level?)

Fig4C-E: better quality panels will be appreciated.

Discussion:

Line 412-413: The results shown in this study are diameter of anterior-posterior midgut junction.

Correlation between this and whole size of the posterior midgut (justification of using one parameter as a reference for a complex 3-dimensional organ) is not clarified in the manuscript.

Discussion would perhaps be better including other factors (than JH) transferred from males by mating that they may act with JH on maturation of the midgut.

We would like to thank all reviewers for their valuable assessment of our work. Here we address each comment in a point-by-point manner:

Reviewer #1 (Remarks to the Author):

Review Commsbio Taracena et al.

In the manuscript, 'Aedes aegypti midgut microbiota establishment in adult females is regulated by Juvenile Hormone to enhance fecundity and fitness', Mabel Taracena and colleagues describe the interesting finding that in the mosquito intestine Juvenile Hormone suppresses the immune response e.g. Anti-Microbial Peptides. JH is known to be boosted upon mating in fruit flies and Taracena et al. discover that the expansion of epithelial tissue leads to more egg production and life expectation, which is very interesting as the observed downregulation of antimicrobial peptides might also cause opposing effects. Overall, the manuscript is well and comprehensively written and data presentation is clear and sound and includes novel findings. However, I have some major and minor remarks concerning the current version of the manuscript, as it is preliminary/incomplete in its writing in the final paragraphs of e.g. results and one method.

We would like to thank the reviewer for the recognition of our work. We have addressed to the comments as follows:

Major:

- Page 12, Line 24-27. Fig.2D The increase in apoptotic signaling is a very interesting finding. Similar observations have been made in the Drosophila midgut, where Notch and EGFR signaling control progenitor apoptosis to balance ISC cell production with actual epithelial cell need (Reiff et al., 2019). As Aedes does not offer the genetic tools for a study of signaling processes with cellular resolution, the authors might have a look whether actual caspase-3 + cells are found and aim to identify whether these cells are morphologically resembling progenitors or enterocytes. It could also briefly be discussed as PCD during mating induced midgut expansion seems counterintuitive.

*As per the reviewer's suggestion, we sought to identify activated caspases in the midgut epithelium of the mosquitoes. For this purpose, we used CellEvent Caspase 3/7 detection reagents from ThermoFisher Scientific. We were able to quantify the number of cells in all groups, and our results show that the mated group does have significantly higher counts than the virgin counterparts (results now added in **Figure 2, as panels D and F**). As per the morphology of the cells, Caspase 3/7 positive cells seemed to be located at mid-height of the midgut epithelium and be either diploid or tetraploid. However, the number of cells identified was not large enough (we counted up to 10 positive cells per gut) to make generalizations about the cell type, and therefore, we have left this identification for further study in future publications. A discussion about PCD during mating while there is a tissue expansion has been added on **Page 18, Line 418**.*

- Page 12, Line 24-27. Fig.2D It should also be noted that expression of a caspase does not equal its activation. The Baena-Lopez lab also revealed non-apoptotic roles for caspases in midgut homeostasis that should be discussed.

We would like to thank the reviewer for bringing this important distinction to our attention. We have modified the figure accordingly and have added a line on the topic to include this point on page 18, line 424.

- Page 15+16 the text does not refer to any panel of Fig.4 apart from 4A. This part definitely needs rework. From an experimental side of view, its hard to see in the 4B whether there is differences, which is better in 4C-E.

All panels from Figure 4 are referenced in the text on Pages 14 and 15, and the figure has been modified to improve its clarity.

- Page 15 to 16, Line 6 following is incomplete

This was a formatting mistake, it is now corrected.

- From M&M and the main text, it does not get entirely clear how 16s as proxy for bacterial number for midgut microbiota is calculated, which raises the next question. Is this referred and normalized to the also observed size increase? So actually, more microbiota per midgut? Or per midgut volume?

The quantification of 16S rRNA gene copies was performed using qPCR, and the values obtained were normalized to the total cDNA content extracted from the RNA from the midgut samples. This normalization strategy allowed us to account for variations in the mass of tissue and account for size increase. In our revised manuscript, we have added text to incorporate the discussion of this concept on Page 17, line 377.

These clarifications and additions enhance the transparency of our methods and strengthen the discussion regarding the potential implications of midgut changes on the interpretation of 16S rRNA gene quantification.

Minor:

- Page 3, Line 23-25 & Page 19, Line 19-21, actual hemolymph 20HE titer increase upon mating was shown in Zipper et al., 2020.

This article has been added (now reference 24) to reflect adequately prior work done in this subject.

- For clarity in all figures, in quantifications of JH fed VF, that should be clearly visible for the reader without the text, like in Fig.4A. In other figures (like Fig2) it just states JH which could be misleading.

The headings of all figures have been formatted so this is no longer confusing and it is clear to the reader that the group was made of virgin females treated with JH.

Reviewer #2 (Remarks to the Author):

The authors showed that mating and JH induce gut growth by stimulating cell proliferation in *Aedes aegypti*. This causes reduction of AMP expression, establishment of gut microbiota population. This finally results in increased reproductive outcome and longer lifespans. Although I agree with the authors' interpretations of the presented data, I see several things are missing to connect all these observations. Please see details below.

We would like to thank the reviewer for pointing out the missing links our manuscript had. We have added the requested information and as a result the data now is more robust. Here we attend the comments point by point:

Major points

- Although it is clear that JH changes the gut size (maturation) by regulating cell proliferation, in order to show this causes reproductive outcomes, it is key to show this leads to a blood meal size, which is not directly shown in this manuscript.

*Thanks to the reviewer's comment, we measured the weights prior to and immediately after engorgement in all groups. While virgin mosquitoes ate approximately 123% of their own body weight in blood, both mated and virgin treated with JH mosquitoes ate \approx 115% of their body weights (Now this result is presented in **Figure 4A, Pages 13-14**). However, there was no significant difference between the net weights of blood consumed amongst all three groups, 3.3 ± 1.1 mg of blood per mosquito in all groups. This information allows us to say that the available biomass of blood for egg production was the same in all groups, and therefore differences in the net count of eggs laid must have a different reason.*

- The other important thing missing in this manuscript is whether mating results in an increase of JH concentration in circulations. Is it possible to look at jhamt and kr-h1 expression in virgin (0 day), 0.5, 1 and 2-day post mating? Since the gut size becomes larger by 48 hours, these four time points must be sufficient to detect expression of these genes if this happens.

As per the reviewer's comments, we searched for validated primers to detect these genes and found suitable sequences for this at Nouzova et al. 2021. We dissected virgin and mated females at 24 hours post-mating. Pools of 10 mosquitoes were made, and the corresponding cDNA was synthesized to assess gene expression by qRT-PCR. However, our results indicate that there were no significant differences in the expression of these two genes between the virgin and mated females. In light of these results, we did not expand our results section to include this data.

Circulating titers of JH in females after mating had already been reported by Salvador Hernandez-Martinez et al., 2015; therefore, we did not repeat these measurements.

- Although I agree mating leads to reduced AMP expression, which results in increased 16S expression, it is not clear how biologically relevant this reduced AMP expression is. It is because AB treatment is not necessarily the same thing. I would like to see whether this reduced AMP expression increases lethality by oral bacterial infection. This immunity-reproduction relation is a classic trade-off that can fundamentally support the authors' conclusion.

*As per the reviewer's suggestion, we promoted an oral infection of an entomopathogenic bacteria, Pseudomonas entomophila, in our three groups: virgin, virgin treated with JH, and mated. As expected, the mated mosquitoes exhibited an increased susceptibility to the infection and saw their survival reduced. However, the virgins treated with JH survived at the same rates as their untreated counterparts. The results have now been added to **Figure 3 as panel D**. This experiment was therefore important for us to observe that upon infection, the immune suppression observed in the JH-treated groups could probably be overruled, different from what happens in the mated group. Studying the molecular pathways involved in this complex regulation was out of the scope of the present manuscript, but it is an interesting finding that we will continue investigating in future directions.*

- It is not clear whether unfertilized eggs by virgin mosquitoes that are treated with JH are qualitatively/quantitatively similar to fertilized eggs. Is it possible to see the size and (if possible) egg contents? I am asking because applying JH may simply induce oviposition of immature eggs.

*We measured egg lengths obtained from blood-fed females from the three groups and found no significant differences between them. These results have now been added to Figure 4 as panel C. Additionally, we analyzed the protein content of the eggs (**Supplemental Figure 3**) and both virgin and virgin treated with JH present the same signature bands, slightly different from the ones observed in the fertilized eggs where the embryogenesis process alters the composition even at early timepoints.*

However, total amounts of protein were calculated from the protein extracts (Page 14, line 325), and these do not show any significant differences between the groups.

Minor points

- Line 94. Typo? Established?

The typo has been corrected.

- Figure 1C, 2B and so on. It is not clear whether the “virgin” means “virgin + acetone” and whether “mated” means “mated = acetone”. It is not clearly written besides the methods. If you compare with JH treated guts, this “virgin” has to be “virgin + acetone” and “mated” has to be “mated + acetone”. Please clarify.

We have opted for labeling the three groups 1) virgin, 2) virgin + JH and 3) mated, and have labeled consistently through the figures. Details about the controls with acetone were left for the methods section alone to simplify the comprehension for the reader throughout the figures.

- Figure 3A, B and C are mixed up in the main text.

The figure has been corrected to match the flow of the text.

- Lines 420-421. Typo. “we” twice.

The typo has been corrected.

Reviewer #3 (Remarks to the Author):

The manuscript titled “Aedes aegypti midgut microbiota establishment in adult females is regulated by Juvenile Hormone to enhance fecundity and fitness” by Taracena et al. presents effects on the female midgut caused by mating and artificial application of JH (a component of materials transferred from male by mating). The authors examined various aspects of effect of mating and JH utilizing a wide range of techniques. The maturation of the midgut prior to the blood meal has long been overlooked and this study sheds light on the potentially critical part of female *Ae. aegypti* physiology. The results presented in the manuscript are new and of interest of the research community. However, the manuscript needs to improve with validation of used techniques, a better organization, and ordered presentation to be convincing. The manuscript would attract more interest with discussion of other factors transferred by mating than JH, which seem also play important roles in midgut maturation. A brief mention on the negative results (attacin) could also support the midgut undergoes complex modulation of gene expression during maturation triggered by mating.

In addition, I would appreciate that authors would have been a little more careful in preparing the manuscript as many points I found could have been caught by themselves.

We would like to thank the reviewer for the careful examination of the work and text, we attended to all the different points raised and as a result the paper is now more detailed and clear.

Points:

Title:

It appears that JH is not only the factor for microbiota establishment from the results, and the title may be misleading.

*A revised title has been included in the manuscript: Juvenile Hormone as a Contributing Factor in Establishing Midgut Microbiota for Fecundity and Fitness Enhancement in Adult Female *Aedes aegypti*.*

Abstract:

I feel it needs more background (why it's done, what questions asked) as it is mostly results and conclusion.

*We appreciate how adding the questions asked in the study can improve how the abstract reads, and we have reviewed the text and added a sentence to this effect: "Additionally, investigating the factors influencing fecundity and longevity is essential, as these parameters significantly impact the mosquitoes' vectorial capacity. In this study, we aimed to address how does mating affect midgut growth in *Aedes aegypti* mosquitoes, what role does Juvenile Hormone (JH) play in this process, and how does it impact the mosquito's immune response and microbiota."*

Line 37: "vector capacity" vectorial capacity?

The typo has been corrected.

Introduction:

Line 60-61: To be accurate, mosquitoes do not transmit diseases, but pathogens.

The sentence has been edited for accuracy.

Line 66: "its development" Do mosquitoes need blood for development? Or authors mean they need blood for development of oocytes?

The sentence has been edited for accuracy.

Line 78: "this tissue" which tissue (fat body or ovary)?

The sentence has been edited to make sure it is the fat body.

Line 94-95: “it is well established that newly emerged mosquitoes do not feed on blood” If so, references needed.

This sentence has been modified for accuracy and a reference has been added (no. 43).

Line 111-112: “As increased fecundity and longevity after mating had already been described in this organism” requires references.

Reference no. 45 has been added.

Materials and Methods:

Line 123: “A. aegypti” please be consistent throughout the manuscript: Ae. aegypti or A. aegypti.

Ae. Aegypti is now consistently used throughout the manuscript.

Line 126-132: This study used two different strains of Ae. aegypti and the experiments were conducted in two different laboratories. In the methods, results and discussion, no clear description of which experiments were conducted with which strain/lab, or if the same experiments were conducted in both strain/lab. I think that, at least, clear statement of which strain/lab was used for which experiment is needed to avoid future confusion. If they observed differences of results by strains, that may also be of interest of researchers in the same field. It is ideal to have justification of use of two different strains for the study.

*The paragraph on mosquito rearing in material and methods has been edited to include the description of which experiments were conducted on which strain/lab for clarity of the methods, **Page 20**. In all the figures it has now been clarified which results pertain to which strain to improve transparency and facilitate for the reader to follow on which experiments are of which strain.*

Line 135: “sheep blood” is it the same “defibrinated sheep blood” appears later in the manuscript?

Defibrinated sheep blood was the correct term, and the text has been modified to reflect this.

Line 146-149: Please provide reference(s) if this method has been previously established (with verification of free of cultivable bacteria). Otherwise, authors may need to mention that they have verified that there is no cultivable bacteria in the gut (with description of its method).

*A reference for the method has been provided (reference no. 64), and the mention of the verification for cultivable bacteria can be found on **Page 22, lines 521-525**.*

Results:

Liner 250-253: Authors did not mention on the establishment of anterior-posterior midgut junction for

the size of the (entire) midgut as it is so mentioned in the legend. Reference 46 does not seem to describe this. If it has been done previously, please cite.

Edits have been made to the lines to improve clarity. The mentioned reference, Park and Shahabuddin 2000, describes the structural organization of the posterior midgut muscles and that is why it is cited upon the mention of the structure of the muscle in the methods and legend but not in the results, to clarify the terminology used to refer to the midgut anatomy to set up the point of reference selected to measure. On the revised version this point is made clear.

Line 256, 257: are the numbers mean \pm SD?

*The paragraph has been edited to clarify that the numbers are mean and SD, **Page 6, line 136.***

Fig1A: Red letters are difficult to see. Both panels seem to have another faint red lines, which are confusing.

The faint red lines have been substituted by black lines and the "J" standing for "junction" has been made black for consistency. The red letters have been placed into a box to make them more visible.

Fig1B: Lacks the units on x-axis and the significance notation (asterisks etc.).

Significance notation has been added. Unit on the x-axis is days after emergence and has been added, as well as the significance notation.

Fig1C: no description of the numbers shown in the figure (mean, or median?). No explanation of dots, lines, boxes, and single * are given.

*Description of the numbers, lines and significance notation has been added to the legend, **Page 7.***

Line 262-269: Not sure why this is here. It seems discussion about following results (line 283-289), and sounds odd here.

Changes in this paragraph have been made to improve the flow of this text.

Line 298-299: seeing the Fig2B, there may also be other factors transferred by mating contributing the phenomenon, and authors did not test Delta expression and caspase 16 for JH-treated virgins.

*We measured gene expression of Delta and Caspase16 in JH-treated virgin females (shown now in **Figure 2 panels E and F**) and found that it was no different than the one found in control virgin females. This, coupled with the new experiments suggested by Reviewer no.1, further supports the notion that other factors transferred during mating contribute to the phenomenon and JH alone is not sufficient to induce the full response observed after mating (**Page 12, line 274**). Now the title and the text throughout the manuscript follow this observation.*

Fig2: as for Fig1 no explanation of dots, lines, boxes are given. Only ** is explained, though *, ***, and ns are also shown.

Description of the numbers, lines and significance notation has been added.

Fig2C-D: relative expression to What? What are a horizontal line across the panels at y=1?

*As the basal or initial state of all groups of adult females is the virgin condition, this group was the one used as control and therefore all horizontal lines at y=1 correspond to the average expression in this group. This has been added to the methods section, **Page 22, lines 521-525**.*

Line 322-323: Fig3A is 16S quantity, not AMPs. (looking at Fig3C) significant difference seems to be only between virgin and JH (they seem to lack significance notation between virgin and mated).

The order of the figure has been corrected and significance notation has been added.

Line 323-326: Fig3B is 16S, and Fig3C is AMP.

The figure has been corrected to match the flow in the text.

Supplemental figure 1 legend:

Line 667: what is "d ()"?

This typo has been deleted from the legend.

SFig1A contains a scale bar, but no description of the size of the bar is given.

Scale bar size has been added to the legend.

Line 336-339: Assuming (the current) Fig3A shows 16S abundance (expression?) relative to host S7 (or actin), the change (expansion) ratio between 2 and 4 or 6 DPE for both virgin and mated groups are

similar (thus virgin group's bacteria also expanded at similar rate to mated group *calculation below). So it looks, to me, that the cell proliferation (and effect of JH) may have more correlation to abundance of bacteria, rather than expansion.

*Ratio calculation (from current Fig3A):

4 vs 2 DPE: virgin: $4.8/2.0 = 2.4$; mated: $425.3/112.4 = 3.8$

6 vs 2 DPE: virgin: $2.0/2.0 = 1$; mated: $128.3/112.4 = 1.14$

The reviewer is correct; with our results, we can report an increase in bacteria abundance, and this has been corrected in the manuscript (Page 3, line 10, Page 16, line 375, and Page 17, line 382).

Fig3: the figures do not match either legends or the text. The reference(s) of relative expression are not clarified. No explanation of bars, dots and significance notations (but ***) are given. What is the horizontal line across the panels at $y=1$?

The figure and the text have been corrected, and explanations for the significance notations have been added.

Fig3: No discussion about attacin are found in the results or discussion.

Discussion about Attacin can now be found on Page 18, line 403.

Fig4A: no explanation of dots, box with whisker, and significance notation is given in legend.

Explanations for these have been added.

Fig4B: missing significance notation (between which are/is significantly different at what level?)

Significance notation has been added.

Fig4C-E: better quality panels will be appreciated.

Figure has been reformatted to incorporate new results and improve quality.

Discussion:

Line 412-413: The results shown in this study are diameter of anterior-posterior midgut junction.

Correlation between this and whole size of the posterior midgut (justification of using one parameter as a reference for a complex 3-dimensional organ) is not clarified in the manuscript.

*In Drosophila, the diameter of the midgut part adjacent to the hindgut (which could be called the “posterior”) had been used previously to infer the size of the organ by Reiff et al 2015. In mosquitoes, the midgut has a very clear division of anterior and posterior, with functional differences between these. The posterior midgut expands to accommodate an incredibly large amount of blood, and therefore, we considered it important to obtain measurements of this organ at the critical point for food storage. The shape of the mosquito posterior midgut, which resembles a larger sac rather than a tube as the anterior midgut, is harder to mount without any wrinkles or twists that could introduce noise in the length measurements. Therefore, measuring the midgut at the anterior to the posterior junction is the best proxy we could find for the full measurement of the gut dimensions. This is now part of the methods section on **Page 23** of the manuscript.*

Discussion would perhaps be better including other factors (than JH) transferred from males by mating that they may act with JH on maturation of the midgut.

*We have included now into the discussion the consideration of other factors transferred from males in the seminal fluid, **Page 17, line 389**. However, an in-depth analysis of these is outside the scope of our attention for this particular manuscript. As there are hundreds of molecules in the seminal fluid, and in the current state of literature only some of these have been fully characterized, it was our opinion that bringing in discussion about this topic would still have many question marks and could be too speculative. More research is clearly needed on this important topic that this reviewer has called attention to.*

REVIEWERS' COMMENTS:

Reviewer #1 (Remarks to the Author):

In this revised manuscript, Taracena et al. addressed my suggestions and concerns raised during the reviewing process. I congratulate all authors for this valuable contribution to the field of endocrine regulation.

Reviewer #2 (Remarks to the Author):

Although the data newly presented in the revised manuscript was not fully positive, the authors have done all the experiments I suggested and carefully rephrased the manuscript based on what we observed. Therefore, I am convinced for publication.

Reviewer #3 (Remarks to the Author):

Overall:

This revised manuscript improved substantially from the initial manuscript by addressing the concerns I pointed and adding additional experiments and data. The paper is scientifically fine as a new knowledge for the community. If I have to say my concern, however, some editorial change may help improve this paper (briefly mention in the points below).

Points:

Results section reads like results and discussion combined (as final paragraph sounded like final statement of entire study). I understand it is tempting to mention interpretations of the observed results where they are presented, but it eliminates what should be in the Discussion.

Figure legends read like results with legends. Yes, interpretation may be useful for the readers, but stating it may be better in the text (my opinion).

Discussion seemed to be a short review for the related fields as if mixture of what should be in the introduction and something may be in the discussion. If this is written to focus recapping readers what this study found, to explain what they mean, and to guide what the future directions are, this section could help readers how important this study is (was).

Line 82: ref 13-18 include Anopheles papers, did they study *Ae. aegypti* as well?

Figure 1 legend: which test used? ANOVA or T-test?

Line 161: Please spell out "ISC".

Line 239-241: unable to find corresponding figure (Referring to Supp Fig 2A, showing 16S for antibiotics)

Fig 4: legend for panel A is missing and that for B-H is in A-F.

Line 311-313: Then what is ** in the figure (4A)? Maybe reflecting initial weights of mosquitoes? Different presentation may be better (?)

Methods: please clarify at what time point after application of JH each experiment was performed.

Here we address each comment in a point-by-point manner:

Reviewer #1 (Remarks to the Author):

In this revised manuscript, Taracena et al. addressed my suggestions and concerns raised during the reviewing process. I congratulate all authors for this valuable contribution to the field of endocrine regulation.

We would like to thank Reviewer #1 for the helpful suggestions and commentary.

Reviewer #2 (Remarks to the Author):

Although the data newly presented in the revised manuscript was not fully positive, the authors have done all the experiments I suggested and carefully rephrased the manuscript based on what we observed. Therefore, I am convinced for publication.

We would like to thank Reviewer #2 for the helpful suggestions and commentary.

Reviewer #3 (Remarks to the Author):

Overall:

This revised manuscript improved substantially from the initial manuscript by addressing the concerns I pointed and adding additional experiments and data. The paper is scientifically fine as a new knowledge for the community. If I have to say my concern, however, some editorial change may help improve this paper (briefly mention in the points below).

We would like to thank Reviewer #3 for the helpful suggestions and we address the comments made here in a point by point manner:

Points:

Results section reads like results and discussion combined (as final paragraph sounded like final statement of entire study). I understand it is tempting to mention interpretations of the observed results where they are presented, but it eliminates what should be in the Discussion.

We acknowledge that extended interpretations of the results should be presented in the discussion section. However, the presentation of basic statements of data interpretation within the results section can aid the reader in understanding the results and, therefore, can facilitate the comprehension of them. In the writing, it can also be considered a matter of style, and it is our view that, in this case, it helps the reader to follow the flow of the results presented. Considering the fact that the other two reviewers didn't see any problem with the overall flow of the article, we have left the core of the result sections as it was. However, we do agree that this should be kept to a minimum, so we have edited the legends.

Figure legends read like results with legends. Yes, interpretation may be useful for the readers, but stating it may be better in the text (my opinion).

As per the reviewer's comments, we revised our legends and found that the legend of Figure 3 contained some statements that were interpretations of results and were included in the main results text. These were cut out to benefit the clarity of this legend. We have modified the legend to remove these sentences.

Discussion seemed to be a short review for the related fields as if mixture of what should be in the introduction and something may be in the discussion. If this is written to focus recapping readers what this study found, to explain what they mean, and to guide what the future directions are, this section could help readers how important this study is (was).

It is very important to have a clear distinction between results and discussion, and on that point, we agree with the reviewer. However, we do not believe that our discussion is written as a short review of the related fields. For each point, we have included background information on what was known in the literature and what our results are; in some cases, we discuss the limitations and specific considerations that pertain to them and the possible implications. We understand that the way a discussion is organized can be a matter of style and that the reviewer may prefer a different one but in this case, we disagree with this assessment of our discussion and would intend to keep our current organization of the manuscript.

Line 82: ref 13-18 include Anopheles papers, did they study Ae. aegypti as well?

The reviewer is correct in pointing this out: 2 of these papers only studied Anopheles, and therefore, the sentence in the main text should not be limited to Ae. Aegypti, and the subject has been changed in the sentence to read as "mosquitoes" to include both species.

Figure 1 legend: which test used? ANOVA or T-test?

In this legend, it is indicated that ANOVA is used in figure 1B and T-test is used in figure 1C.

Line 161: Please spell out "ISC".

Intestinal stem cell marker has been spelled out in line 161.

Line 239-241: unable to find corresponding figure (Referring to Supp Fig 2A, showing 16S for antibiotics)

We had missed this detail; the point mentioned by the reviewer here corresponds to the assay counting Ph3-positive cells in virgin females fed or not with antibiotics. This figure was indeed missing from Supp Fig 1. We have added it and modified the main text (line 199) and the legend accordingly.

Fig 4: legend for panel A is missing and that for B-H is in A-F.

We would like to thank the reviewer for catching this. As a matter of fact, the legend for 1C was also missing and we have corrected this in the revised version of the manuscript.

Line 311-313: Then what is ** in the figure (4A)? Maybe reflecting initial weights of mosquitoes?
Different presentation may be better (?)

The figure presents the data for the weights of whole-body mosquitoes in the two feeding conditions. As the sugar-fed condition is different in weight itself, this basal difference seems to impact the total body weight of the mosquitos' post-blood meal. The sentence mentioned to the reviewer refers to the actual amount of blood ingested, calculated by the subtraction of the pre-blood-feeding weight. It does not make sense to present it differently or separately as it is calculated out of the average of the weights per group.

Methods: please clarify at what time point after application of JH each experiment was performed.

The information has been added to the methods section, line 410.